# Methodology for the Localization of Wool Collecting Centers: The Case Study of Sicily

Mariaconcetta Ganci, Luisa Biondi *, Monica C. M. Parlato  and Simona M. C. Porto 

Department of Agriculture, Food and Environment, University of Catania, Via Santa Sofia 100,
95123 Catania, Italy
* Correspondence: luisa.biondi@unict.it

**Abstract:** Sustainable rural development depends on a multidimensional process based on the interaction of the economic, cultural, social, environmental, and political subsystems. Many studies have ascribed the extensive livestock systems a potential role in sustainable rural development. Sheep farming in Europe involves meat or dairy breeds that produce coarse wool unsuitable for textile use. Coarse wool has been historically used for mattresses or pillows; nowadays, it is a farm waste. The aim of this study was to suggest a methodology for the localization of wool collecting centers within a study area, i.e., the Sicily Region, in order to valorize a livestock waste, i.e., sheep wool. This methodology is based on a detailed database of the local livestock population and GIS usage. Moreover, this method could be replicable in the whole EU territory, where the EC regulation n. 21/2004 has established that each Member State set up a system for the identification and registration of ovine animals, for traceability veterinary reasons. On the basis of the number of animals shorn on a municipal basis, and the existing main roads, the most suitable areas for the localization of four wool collection centers have been identified. Furthermore, it was also hypothesized that each collecting center could be specialized in a specific treatment for subsequent wool use (e.g., amendment fertilizers, keratin extraction, green building material or geo-textile production).

**Keywords:** rural sustainability; raw sheep wool; valorization of livestock waste; circular economy

## 1. Introduction

The sustainability of rural areas is strongly affected by land abandonment, a consequence of the migration process of the rural population towards urban centers. In 2015, about 54% of the world's population was urban, but in 2050 more than two-thirds of all people may be living in urban areas [1]. This phenomenon will have a high impact on the landscape and bio-system. In the EU, the European Landscape Convention (2000), the EU Habitats Directive, and Agenda 2000 seek to fight rural area abandonment by the preservation and valorization of traditional rural landscapes. Sustainable rural development aims to increase the quality of life of the residents, and their active participation in social and cultural life. It is referred to as a multidimensional process based on the interaction of the economic, cultural, social, environmental, and political subsystems, as per the three pillars of sustainability. In this context, several studies have proposed organic livestock systems have a potential role in sustainable rural development [2].

Small ruminants are very important for food security; they are efficient, suitable for organic systems, and can adapt to different climatic conditions: about 56% of the world's small ruminants are in arid zones, and 27%, and 21% in temperate and humid zones, respectively [3]. The sheep world population is around 1176 million, with China breeding the highest number (about 15% of the total) [4]. About 129 million sheep live in Europe [4], representing 8% of the total world sheep population. In the northern and central European countries, meat sheep breeds are prevailing, differently from in southern countries, where dairy sheep breeds represent most of the sheep population. In Italy, in particular, dairy sheep breeds represent more than 65% of the Italian sheep population (6 million in 2021) [5].

On a global scale, in 2020, annual raw wool production was about 1.8 million tons [6]. About 60% of world wool production comes from three countries, Australia, China, and New Zealand, in descending order [6]. Wool wastes, all over the world, come from different steps of the wool-production chain: in the farms, as raw wool from sheared animals when wool is not suitable for textile use, or in the textile factories as fibers and leftover fabric wastes. However, the farms' raw wool represents the main issue, because it is at the beginning of the production chain [7].

European sheep breeds have been selected for meat or milk production. These genotypes produce a coarse wool of low textile quality. In dairy (e.g., the Italian *Valle del Belice*), meat (e.g., the Dutch Texel) and wool (e.g., Merino) breeds, the fiber diameter is, respectively, 70 μm [8], 32–40 μm [9] and 17–22 μm [10]. The literature shows a negative relationship between fiber diameter and mechanical strength, which makes the thicker wool fiber unsuitable for the textile industry. Parlato et al. (2022a) [8] observed, in the *Valle del Belice*, that wool fiber has a very low mechanical strength value (137 MPa) as compared to the average value (250 MPa) observed in the wool textile industry (e.g., Merino or Romney wool). Mean fiber diameter is by far the most important physical property affecting processing performance, fabric properties, consumer evaluation, and price per kilogram [11].

Coarse wool, typically produced by dairy sheep breeds, has historically been a valuable commodity, mainly used for mattresses or pillows; however, recently, it has become a farm waste. Based on the EC Regulation 1069 (2009) and EU Regulation 142 (2011), wool is classified as a category 3 animal by-product (ABP), which means low-risk waste [12]. The growing awareness of environmental pollution and the increasing need for safe and sustainable biological materials represent the drivers in the search of methods for recycling wool waste [13,14] according to a circular economy approach. The literature shows several alternative uses of coarse raw wool, which make non-textile wool a valuable source: green building material [15,16], amendment fertilizers [14], geo-textile production [17], bio-gas production [18], heavy metals-polluted water treatment [7], oil absorbance in sea disasters [7,19] and keratin extraction [7]. In particular, wool can replace plastic components in the green building and geotextile sectors. Considering the growing concern about microplastics pollution in water and soil, this particular use of non-textile wool can play an important role around the world. However, before its usage, wool needs to be chemically, physically, or enzymatically treated [14]. In the EU, any pre-treatment of waste wool can be performed in specialized and authorized centers exclusively (EU regulations n. 1069/2009, n. 142/2011, n. 1063/2012 and n. 1097/2012).

The aim of this study was to provide a method for the localization of wool collecting centers in a given geographic area. The pillars of this method are a detailed database of the local livestock population and GIS usage. This method can be replicated wherever an open access database on sheep farms' locations and population has been implemented, such as throughout the EU. Indeed, the EC regulation n. 21/2004 has required that each Member State set up a system for the identification and registration of ovine and caprine animals for traceability reasons; as from 2008, for each animal, the genotype is also to be considered. Furthermore, the method was developed for a non-specialized GIS user.

## 2. Materials and Methods

### 2.1. Materials

#### 2.1.1. Wool

Wool is a keratinic fiber containing carbon (50%), oxygen (22–24%), nitrogen (16–17%), hydrogen (7%) and sulfur (3–4%). Each fiber is formed by an external horny cuticle, which, upon observation under a microscope, appears to be formed by a series of flakes or scales (Figure 1a). Below, separated by a protective sheath called the elasticum, there is a fibrous tissue consisting of fusiform cells (fibrils) cemented by a connective tissue; inside, there is the medullary canal formed by air-filled rhombic cells, which are very small or completely absent in the finest wools [20] (Figure 1b).

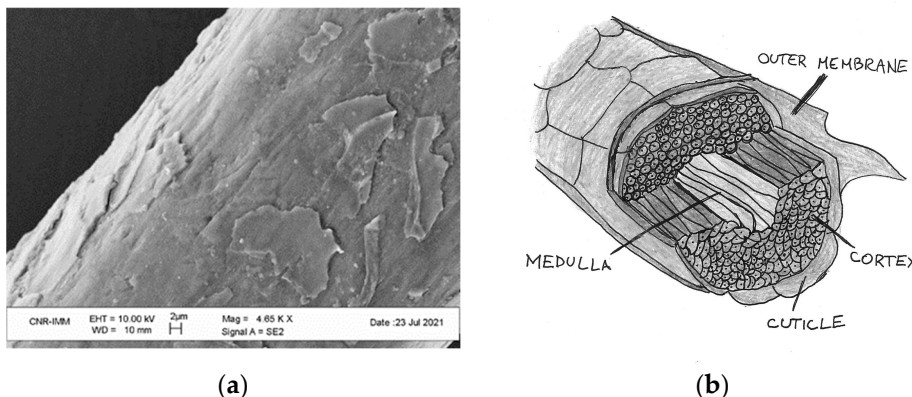

**(a)**                           **(b)**

**Figure 1.** (**a**) Scanning electron micrograph of a *Valle del Belice* fiber, showing overlapping cuticle cells. (**b**) Structure of a wool fiber.

The specific weight of wool ranges 18–20 kg m$^{-3}$ [21]; the specific thermal capacity ($c_p$) of wool is between 1600 and 1700 J kg$^{-1}$ K$^{-1}$ [21], the thermal conductivity ($\lambda$) varies between 0.034 and 0.067 (W m$^{-1}$ K$^{-1}$) [15]; wool is a hot fiber at touch, and is suitable for thermal insulation. Among textile fibers, wool is the least tenacious (with 1.2 to 1.7 g/den); on the contrary, wool exhibits a high elongation capacity (30–50%); its wear resistance is the highest among all textile fibers, and in fact, it is possible to regenerate it for at least two work cycles (regenerated wool). One of the most important characteristics of wool is its hygroscopicity: it can absorb 17% moisture without the fiber undergoing any external change, and 50% without deforming. The wool fibers are characterized by a curl that tends to take on a helical shape, with a greater number of undulations in the finest types: it gives elasticity and cohesion to the fibers, favoring the production of yarns; it improves wear resistance, bulkiness, thermal insulation, resilience, and non-deformability. Moist heat, reducing the curl, diminishes these qualities. Wool effectively resists acids, while it is particularly sensitive, especially when hot, to alkalis; under light, wool turns yellow and weak; its resistance to mold and bacteria is the best among the natural fibers, but it is susceptible to attack by moths [20].

### 2.1.2. Study Area

The study area is Sicily, the greatest among the Mediterranean Sea islands, surrounded by the Tyrrhenian Sea to the North, the Ionian Sea to the East and the Mediterranean on the remaining coasts. It extends for 25,707 km$^2$ and it is the largest Italian region. The Sicilian territory is divided into 9 provinces (aggregates of neighbouring municipalities), and mainly consists of hills (61%) and mountains (25%); the remaining 14% of the territory consists of plains. In Sicily, there are several marginal areas where extensive livestock systems, including sheep farms, are located. Sicily is the second Italian region in terms of sheep population, after Sardinia [5].

### 2.2. Methodology

In this paper, a four-step methodology to achieve the proposed aim, the localization of collection centers for raw wool within a study area, is suggested. In Figure 2, the flow chart of the proposed methodology is shown.

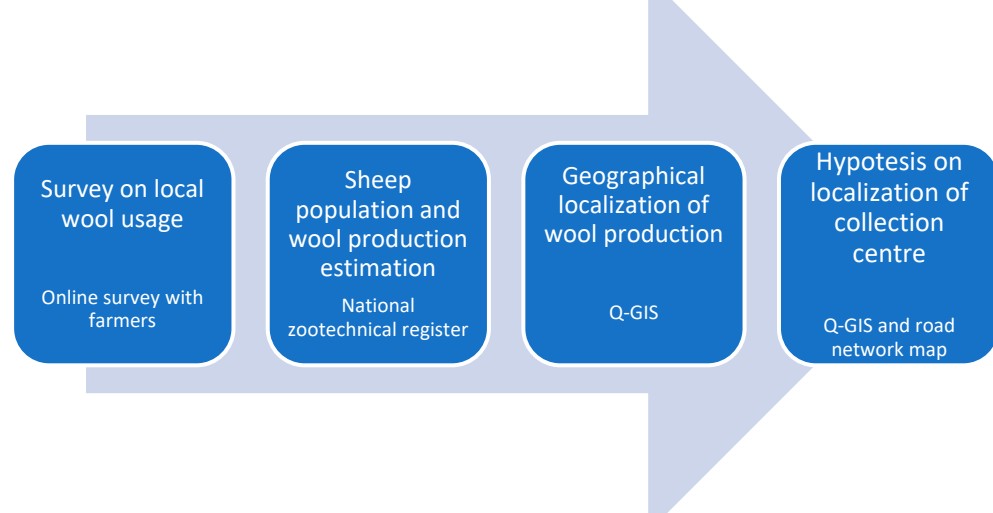

**Figure 2.** Flowchart of the methodology.

### 2.2.1. Survey on Local Wool Usage

The survey was launched online by proposing an anonymous questionnaire to farmers, filled on a voluntary basis. The questionnaire was created on the Google forms platform (https://www.google.it/intl/it/forms/about/) on 3 May 2021; it was disseminated through Facebook and WhatsApp from 12 May to 30 June 2021.

After generic geographical information (municipality and province) on farm localization was collected, the questionnaire contained 29 questions split into two sections: production and disposal logistics. The "production" section contained questions on the number of animals (sheep and other species, if any), breeds and genotypes, products sold and their relative weights in terms of income, numbers of sheep shorn in the period between 2018 and 2021, and wool production. The "disposal logistics" section contained questions on wool usage, shearing costs and incomes from wool, means of transport and distances to disposal centers, means of disposal in the last 4 years, knowledge about alternative destinations of coarse wool, and interest in the creation of a center for wool evaluation in the territory.

### 2.2.2. Sheep Population and Wool Production Estimation

Wool production has been estimated using (i) sheep population data in Sicily and (ii) individual wool production levels in different breeds and sexes.

The sheep population in Sicily (reference date 30 June 2021) was obtained by processing the data, which is freely accessible by a generic user, shown in the database of the National Zootechnical Registry (Banca Dati Nazionale dell'Anagrafe Zootecnica (BDN)) [5]. The BDN shows data on the number of sheep in each municipality; within each municipality, the number of animals of each breed and sex is shown; within each breed and sex, the animals are split into classes of age (0–12 months; 1–2 years; 2–3 years; 3–5 years; 5–8 years; 8–12 years; higher than 12 years).

Wool production within each municipality and within each of the nine Sicilian provinces was estimated by counting the shorn animals separately for each breed and sex within breed. Young animals (less than 12 months of age) were not considered for wool production because they are not shorn. As for the breed of the shorn animals considered for the wool production estimation, all the autochthonous Sicilian breeds were included, regardless of their numerical consistency. All the other breeds registered in the BDN were considered only in cases when their regional consistency was higher than 1% of the total Sicilian sheep population.

The number of shorn animals, according to their genetic origin and sex, was then multiplied by the individual yearly wool production level. Table 1 shows the number of shorn animals (sum of rams and ewes) to account for wool production estimation and the literature data for individual wool production (kg/head) in the different breeds.

**Table 1.** Total number of shorn animals and average annual level of wool production in sheep breeds reared in Sicily.

| Breed | N° of Shorn Animals (Both Sexes) | Wool Production (kg/Head) | | |
|---|---|---|---|---|
| | | Ram [22] | Ewe [22] | Average |
| *Barbaresca* | 875 | 6.5 | 3.1 | 4.8 |
| *Comisana* | 4013 | 2.5 | 1.3 | 1.9 |
| *Noticiana* (a) | 355 | 2.5 | 1.3 | 1.9 |
| *Pinzirita* | 5825 | 2.5 | 1.6 | 2.05 |
| *Sarda* | 8555 | 2.5 | 1.1 | 1.8 |
| *Valle del Belice* | 128,605 | 2.2 | 1.5 | 1.85 |
| Crossbred (b) | 491,520 | 2.3 | 1.5 | 1.9 |

(a) Autochtonous breed originating, from the *Comisana* breed, in the Siracusa area. *Comisana* breed data have been considered because the literature does not show wool production data. (b) Wool production data for rams and ewes have been estimated as weighted average value of annual wool production per head in the *Barbaresca*, *Comisana*, *Pinzirita*, *Sarda*, and *Valle del Belice* breeds.

### 2.2.3. Geographical Localization of Wool Production

To obtain information about the geographical areas where sheep population and wool production are located, a Geographical Information System model was implemented and applied. The open source QGIS software (version 3.1) and the regional administrative boundaries were used to develop the maps. The map of the administrative boundaries was downloaded from the *Geoportale Nazionale* [23] through the Web Coverage Service (WCS); this map includes both municipal and provincial boundaries. For the thematic map showing the localization of sheep farms at the provincial level, two maps have been created based on (i) the number of shorn animals and (ii) the percentage contribution to total regional wool production. For the thematic map showing the localization of sheep farms at the municipal level, data of sheep shorn in each municipality have been downloaded from the GIS system. Three classes have been identified on the basis of the number of shorn sheep: class I: < 3500; class II 3500–8000; class III > 8000 shorn animals.

### 2.2.4. Localization of Collection Center

The localization of the collection centers was hypothesized considering the following points: (i) wool concentration production at municipal level, on the basis of the three classes of shorn sheep described in Section 2.2.3; (ii) road infrastructure map, including state roads and highways. Additionally, here, a road infrastructure map has been downloaded from the *Geoportale Nazionale* [23] through the Web Coverage Service (WCS).

### 3. Results

*3.1. Survey on Local Wool Usage*

The survey was carried out during the months of May and June 2021. It involved 52 dairy sheep farms, corresponding to about 1% of the total (nearly 8500) Sicilian sheep farms. In total, 38% of the interviewed farmers were located in the province of Enna, around 20% in the province of Palermo, and the remaining ones were equally spread within the other provinces. Despite the small number of answers, a clear idea of the raw wool destination in Sicily has been obtained. The most interesting result is that more than half of the farmers interviewed (around the 54%) were not able to quantify the yearly wool production; indeed, some farmers underestimated the individual wool production of their animals (less than 1 kg/head); some others gave values higher than 3 kg/head. In total,

46% of the farmers, on the other side, declared an average production ranging 1–2 kg per head, in accordance with values found in the literature [22]. As regards the destination of the wool, it emerged that in 52% of cases (27 farms), it remains in the farm and is not used in any way. In 20% of cases (11 farms), farmers declared that they were able to sell the wool, indicating sales prices ranging from EUR 15 to 25 per 100 kg of raw wool. Only three farmers bring the wool to the disposal center and only one of them indicated the cost of disposal, approximately equal to EUR 50 per year. The remaining 11 farmers (20%) did not indicate a precise destination for the wool; one replied that they give it as a gift, one said that they use it in agriculture for mulching or as a fertilizer for the soil, and one admitted to burning it illegally. Almost all the farmers (98.1%) involved in the survey expressed their interest in transferring the wool to a collection center for its enhancement.

### 3.2. Sheep Population Distribution and Geographical Localization of Wool Production

Data from the National Zootechnical Registry (BDN) (June 2021) show that Sicily has a sheep population of about 725,000 heads divided into about 8500 farms, with an average of 85 heads per farm. The number above includes all the genetic types mentioned in the BDN, including foreign breeds, with an incidence of <1% in the total, and young animals (<12 months of age). The distribution of the heads among the nine provinces is not homogeneous: three provinces have a consistency greater than 100 thousand heads (Palermo, Enna, and Agrigento); four provinces a consistency between 50 and 100 thousand heads, and the remaining two provinces a consistency of less than 50 thousand heads (Table 2).

**Table 2.** Number of bred sheep and shorn sheep, and related wool production on a province basis.

| Province | Total Sheep | Shorn Sheep | Raw Wool Production (kg $10^3$) |
|---|---|---|---|
| Agrigento | 105,417 | 92,160 | 139.85 |
| Caltanissetta | 64,464 | 56,575 | 85.44 |
| Catania | 83,018 | 71,818 | 109.67 |
| Enna | 11,299 | 94,438 | 143.97 |
| Messina | 72,976 | 66,403 | 103.29 |
| Palermo | 153,992 | 138,428 | 211.12 |
| Ragusa | 26,011 | 23,770 | 36.14 |
| Syracuse | 30,700 | 27,450 | 41.80 |
| Trapani | 77,184 | 68,706 | 104.39 |
| Total | 725,061 | 639,748 | 975.68 |

As indicated in the Materials and Methods, the number of shorn animals (Table 2) was estimated by processing the data from the BDN, taking into account the wool breed production level of the different breeds. The number of shorn animals includes all Sicilian indigenous breeds (regardless of their incidence in the total sheep population) and the *Sarda* breed, the only non-native breed with an incidence higher than 1% of the total sheep population reared in Sicily. Furthermore, since the sheep are shorn after one year, young animals (less than 12 months, 80,853 heads in total) were excluded. Overall, the estimated number of shorn animals was equal to 639,748, and an annual wool production of almost 980 tons was estimated (Table 2).

By means of the QGIS tool, a thematic map concerning the number of sheep shorn in Sicily on a provincial level was created; it emerged that wool production is mainly concentrated in three provinces, Palermo, Agrigento and Enna (Figure 3).

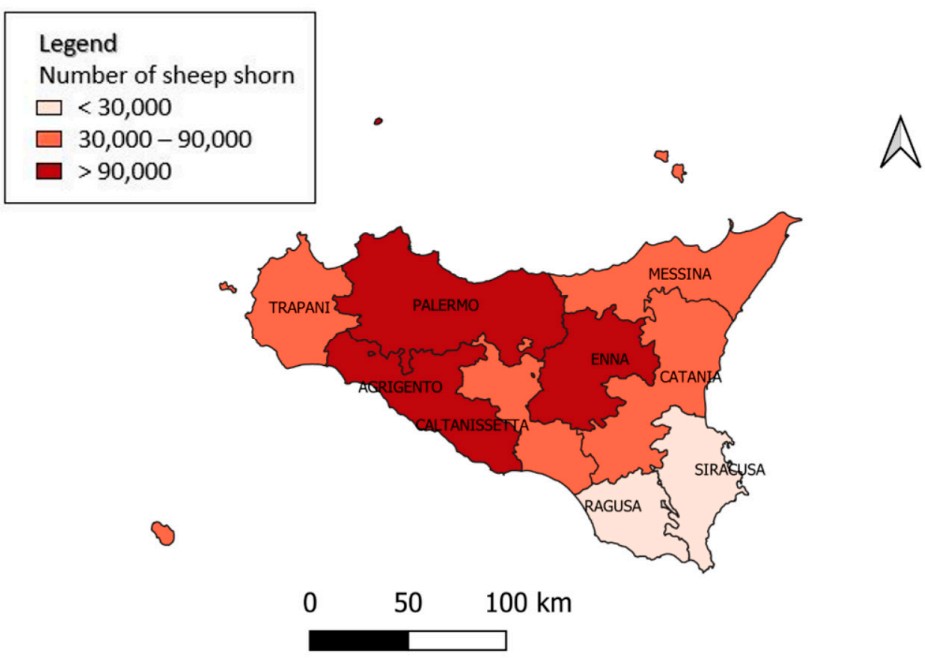

**Figure 3.** Consistency classes of the numbers of sheep shorn in Sicily on a Provincial level.

Data from Table 2 show that the province of Palermo contributes with the highest wool production level, as compared to the other provinces (over 200 10³ kg), corresponding to more than 20% of the total regional raw wool for a year (Figure 4). It is easy to calculate that over the 50% of the Sicilian wool production is concentrated in the three provinces of Palermo, Enna, and Agrigento.

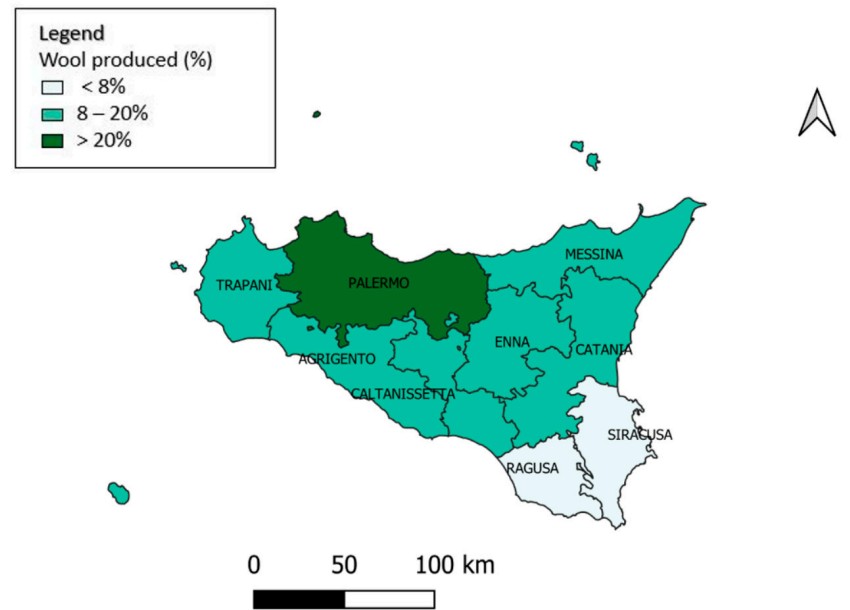

**Figure 4.** Percentage of raw wool yearly produced on a province basis (%).

As for the genetic types raised in Sicily, the processing of the BDN data allows us to calculate that about 75% of the Sicilian sheep population consists of crossbreds, in total about 545,000 head. The *Valle del Belice* sheep is the most farmed in Sicily (over 150,000 head); this breed derives from the crossbred *Pinzirita* x *Comisana*, which is subsequently crossed with the *Sarda* breed [24]. The other breeds present, in decreasing order, are *Sarda* (almost 10,000 heads), *Pinzirita* (6000 heads) and *Comisana* (about 4600 heads). In most of the Si-

cilian provinces, eight out of nine, crossbred and *Valle del Belice* are the most represented genetic types; an exception is the province of Messina, where, besides the crossbreeds, the most common breed is the *Pinzirita* (data not shown). As for the contribution of the genetic types to wool regional production, considering the method of calculating the shorn animals described in the Materials and Methods, 77% of the shorn animals in Sicily are crossbred, 20% are of the *Valle del Belice* breed, and the remaining 3% includes the other breeds.

### 3.3. Localization of Collection Center

The main purpose of this study is to identify a possible location for a collection center for coarse wool valorization. To this end, after the elaboration of data related to wool production at a provincial level, the data at the municipal level were used. As indicated in the Materials and Methods, the municipalities were divided into three categories based on the consistency of the number of shorn heads (<3500; 3500–8000; >8000).

Figure 5 identifies, with different colors, the municipalities with the three classes of shorn sheep. By means of a QGIS tools, in Figure 5, the road infrastructures have been overlapped, showing both statal roads (blue line) and highways (blue tick line).

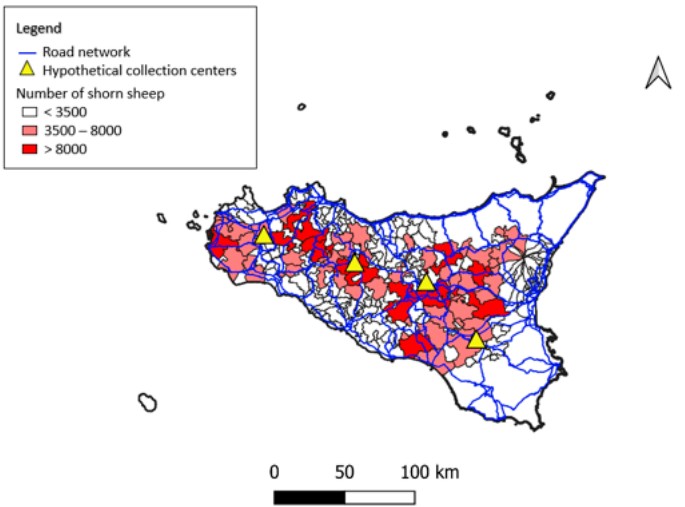

**Figure 5.** Hypothetical location of wool collection centers in relation to the concentration of the numbers of shorn animals on a municipal scale and the main existing roads.

Therefore, based on the consistency of shorn animals in the various municipalities and on the existing road network, four collection centers have been hypothesized, to cover all areas of Sicily.

### 4. Discussion

Sheep breeding, and ruminant breeding in general, represents a tool to counteract marginal area abandonment and improve the sustainability of rural areas. Marginal areas cannot be used as croplands because of the topography, poor soils, climatic conditions, or scarce accessibility of services. In marginal areas, grassland (e.g., the main feed source for ruminants) plays a fundamental role against soil erosion and supports biodiversity preservation. In some Spanish areas, livestock grazing in the forests or shrublands provides an important service for fire prevention [25,26]. Extensive ruminant breeding systems on grasslands reduces the production pressure on the globally limited arable land, and produces food based on local resources [27], thus reducing the feed-to-food competition. Finally, livestock extensive systems also represent a sink of traditional and historical knowledge for cheese or meat-derived niche products, often labeled as PDO (Protected Designation of Origin) or PGI (Protected Geographical Indication).

In Sicily, sheep breeding is aimed at cheese and ricotta cheese production; three PDO cheeses come from this Italian region: Pecorino Siciliano, Vastedda della *Valle del*

*Belice*, and Piacentinu ennese. Ricotta cheese is a by-product of cheese making, with high economic value; it also ensures a daily income to farmers in the period of ewe lactation. Ricotta cheese is traditionally highly appreciated for its multiple culinary uses, from appetizers to sweets. Dairy breeds and crossbred genotypes, however, produce a coarse wool (fiber diameter > 26μm according to the quality classification proposed by Petek et al., 2021 [28]) that currently does not present any economic value, such that the wool represents a farm issue.

The online survey shows the high cost of yearly shearing as compared to wool value. Farmers have indicated a shearing cost per head ranging from EUR 1 to 5, in agreement with the cost of 3 EUR/head suggested by Pagliarino et al., 2016 [29], and a sale price ranging from EUR 15 to 25 per 100 kg for coarse wool. Considering an average wool production level equal to 1.5 kg per head (data of *Valle del Belice* and crossbred animals, which represents 97% of the sheep population in Sicily), it is easy to calculate that in order to obtain 100 kg of wool, a farmer needs to shear 67 animals with an average cost of EUR 201. These data clearly indicate that the cost of shearing exceeds the value of wool. For this reason, almost all the surveyed farmers demonstrated their interest in the hypothetical establishment of a wool collection center aimed at wool value enhancement.

This study represents a methodological deepening of the study by Parlato et al. (2022b) [12]. These authors based their study on the number of farms and the total sheep population on a provincial basis. In the present study, the data on shorn animals on a municipal basis have been considered. In addition, sheep breed and sex (and their respective wool production levels) were also considered. The choice to consider the number of shorn sheep on a municipal basis, i.e., a more detailed territorial level as compared to province, is linked to the low number of sheep per farm; therefore, sheep are spread in a wide area, and the number of farms (or the number of sheep) on a provincial basis could produce misleading results. In each Sicilian farm, an average number of 85 heads are present, as compared to 210 heads per farm in Sardinia, the most important Italian region for sheep breeding [5]. Moreover, in this study, differently from the study by Parlato et al. (2022b) [12], young animals (<1 year) were not considered among the shorn animals, because they are not shorn. On average, in Italian sheep farms, young animals represent about 9% of the total farmed animals [5].

According to the number of shorn animals on a municipal basis, we have hypothesized four different wool collection centers (Figure 5). One could be established in the southeastern area of the Sicilian region, where three state roads meet; this could be used to collect wool from the sheep farms in the provinces of Syracuse, Ragusa, and Catania (southern area). Another center could be located close to the town of Enna, where a highway and two state roads are present. This center could serve farms in Enna, Messina, and in the northern area of the provinces of Catania and Caltanissetta. The third center could be in the province of Agrigento, close to the provincial borders of Caltanissetta and Palermo, served by a state road. Finally, the fourth center could be located between the Palermo and Trapani provinces, where a highway and two state roads are present. As compared to the study by Parlato et al. (2022b) [12], the number of shorn animals and the different territorial levels adopted for wool production estimation produced a more detailed localization of the hypothetical wool collecting centers. In addition, this study implies the opportunity to consider an additional center devoted to the western area of Sicily.

## 5. Conclusions

In this case study, a methodology was developed to recognize suitable areas for the localization of one (or more) collection center, in order to generate a wool chain within the selected area, i.e., the Sicilian region. With the aim of minimizing the impact on the environment, by considering the existing main road infrastructures, the most suitable areas for this purpose, and not an exact location, were identified. In Sicily, the poor existing railway lines (only 37% are electrified and only 12% have double rails) heavily reduce the sustainability of the process of waste wool valorization. Moreover, the existing railway lines

do not meet the requirements of European Regulation corridors and freight services [30,31]. Indeed, in Sicily, about 90% of goods travel by road.

The further development of this study can consider other important aspects in terms of sustainability. The raw wool needs to be washed in order to remove dirtiness and reduce the bacterial load before any alternative usage. The next step before planning collection centers might introduce into the methodology the evaluation of water source availability in the selected areas, and the way to purify and re-use polluted water.

Moreover, the sustainability of the collecting centers concerns several technical and socio-economic aspects. The technical aspect involves farmers' training on hygienic and welfare practices devoted to the production of a "clean" raw wool, considering that fleece dirtiness is an indicator used to assess the hygiene of the lying area [32]. As for the socio-economic aspects, the collecting centers could become a Living Lab of the recently established Sicilian Wool District (Distretto laniero siciliano, founded in 2021), which aims at evaluating sheep wool through a circular economy plan, by exploring the possibility for its reuse in nursery, building or textile sectors [33,34]. Looking also at these complementary services, organizing four collecting centers in Sicily could ensure a widespread presence in the territory. Furthermore, each collection center could specialize in specific treatments for the subsequent use of wool (e.g., amendment fertilizers, keratin extraction, green building material or geo-textile production). The results obtained in this case study could be useful for policy-makers to develop a strategic and sustainable plan for the sheep wool chain in Sicily.

**Author Contributions:** Conceptualization, M.G., L.B., M.C.M.P. and S.M.C.P.; methodology, M.G. and L.B.; software, M.G.; validation, L.B. and M.C.M.P.; formal analysis, M.G.; investigation, M.G.; resources, L.B.; data curation, L.B.; writing—original draft preparation, M.G.; writing—review and editing, L.B. and M.C.M.P.; visualization, L.B. and M.C.M.P.; supervision, S.M.C.P. The study was inspired by the MSc Thesis LA LANA SICILIANA: UNA RISORSA DA VALORIZZARE. Sicilian wool: a resource to give value of M.G., student at the Department of Agriculture, Food and Environment —Di3A—University of Catania (Italy). All authors have read and agreed to the published version of the manuscript.

**Funding:** This research received no external funding.

**Institutional Review Board Statement:** Not applicable.

**Informed Consent Statement:** Not applicable.

**Data Availability Statement:** Not applicable.

**Acknowledgments:** The authors would like to thank all anonymous farmers who responded to the online survey.

**Conflicts of Interest:** The authors declare no conflict of interest.

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
