# Peer review of "Methodology for the Localization of Wool Collecting Centers: The Case Study of Sicily"

_sustainability, doi:10.3390/su141610378_

Round 1

Reviewer 1 Report

It is a very important revision work, but I have some observations/comments that would allow it to be substantially improved.

A detailed and current review should be made of the problem of microplastics and their impact on the environment, assuming that most of them come from clothing or fillers. The fact that contamination is produced mostly from washing machines, from where they cannot be filtered efficiently, generates a potential growth of its effect that can only be stopped by replacing it with natural fibers such as wool. Here is a great challenge for our wool productions all over the planet and above all for the wool that is used for something other than clothing.

The other problem to be defined is the issue of animal welfare associated with sheep management, more than all the shearing that is objected to by animal organizations. This review should include information on the matter, how shearing is done, the antiparasitics used, etc.

With these additions, it is suggested to improve the hypothesis and the consequent objectives set.

Author Response

Dear Editor,

please find below our reply to the reviewer n. 1.

Point 1: A detailed and current review should be made of the problem of microplastics and their impact on the environment, assuming that most of them come from clothing or fillers. The fact that contamination is produced mostly from washing machines, from where they cannot be filtered efficiently, generates a potential growth of its effect that can only be stopped by replacing it with natural fibers such as wool. Here is a great challenge for our wool productions all over the planet and above all for the wool that is used for something other than clothing.

Response point 1: We thanks the reviewer for focusing on this issue. In the introduction of the revised manuscript, we have mentioned the advantage of using wool instead of plastics as an alternative material in green building or in geo-textile production in order to reduce the huge issue of plastics pollution. However, we feel that “a detailed and current review … the problem of microplastics” is out of scope in our introduction. We did not aim at studying or reviewing not-textile wool alternative usage. The core of our paper is to provide a simple and replicable method for the localization of wool collection centres in a given geographic area.

Point 2: The other problem to be defined is the issue of animal welfare associated with sheep management, more than all the shearing that is objected to by animal organizations. This review should include information on the matter, how shearing is done, the antiparasitics used, etc.

Response point 2: In the submitted manuscript, we have indicated that fleece dirtiness is an indicator of the hygienic conditions of the barns and, in general, of animals’ welfare. However, we have mentioned these aspects as an example of potential way to make the wool-collecting centre a multitasking services centre (e.g. enabling training of farmers or providing technical services). We are aware that animals’ welfare is very important in the daily farm management; however, “the issue of animal welfare associated with sheep management” is not the purpose of this study.

We have not carried out any investigation on the perception of sheep shearing by the Sicilian population; furthermore, to the best of our knowledge, there are no published papers on the subject developed in Italy. Consequently, we have no scientific data referring to the study area to be able to “discuss shearing objected to by animal organizations”. Furthermore, this is not the purpose of this study.

Finally, our study i) does not address the scientific effects of shearing on animals (e.g. the stressful effects of shearing, the relationships between shearing and heat tolerance or between shearing and productive and/or reproductive efficiency) and ii) does not deal with shearing management in the study area. Therefore, even though there are a number of published studies on Mediterranean dairy sheep breeds, i.e the sheep typically bred in the study area, we strongly believe that dealing with this subject in the introduction or in the discussion can be misleading.

Point 3: With these additions, it is suggested to improve the hypothesis and the consequent objectives set.

Response point 3: Taking into account the reasons why we believe that the previous points do not fully meet the purpose of our manuscript, we think that "the hypothesis and the consequent objectives" do not require to be modified. Again, we point out that the aim our paper is to provide a simple and replicable method for the localization of wool collection centres in a given geographic area.

Reviewer 2 Report

The article is very interesting and deserves to be published. It has not only scientific but also application strength, perhaps it will give food for thought to people involved in broadly understood agriculture in a given area.

Author Response

We thank the reviewer n. 2 for appreciating our manuscript.

Reviewer 3 Report

Dear authors, the topic is interesting and well structure. Only some suggestions:

Introduction

1.       Line 28, 40, 41: it should be better express percentage with its symbol %

2.       Manuscript: insert space between value and unit

3.       You should state the novelty of you study

Results:

1.       Line 210-212: It should be better if you adopt a unique way to express euros (Line 210) and (Line 212)

Author Response

Dear Editor,

please find below our reply to reviewer n. 2.

Introduction

Point 1: Line 28, 40, 41: it should be better express percentage with its symbol %

DONE

Point 2: Manuscript: insert space between value and unit

DONE

Point 3:  You should state the novelty of you study

In the introduction of the revised manuscript we added the following sentences:

This method can be replicated wherever an open access database on sheep farms locations and population has been implemented, such as throughout the EU.

Furthermore, the method was developed for a non-specialized GIS user.”

Reproducibility and ease of use of the method, although not "new" per se, are the strengths of this study. Our idea is to develop a scientific method that is, at the same time, an easy and useful technical method for territorial planning. We strongly hope the revised introduction meets reviewer’s request.

Results:

Point 1: Line 210-212: It should be better if you adopt a unique way to express euros (Line 210) and € (Line 212)

DONE